# Continuous Particle Aggregation and Separation in Acoustofluidic Microchannels Driven by Standing Lamb Waves

**DOI:** 10.3390/mi13122175

**Published:** 2022-12-08

**Authors:** Jin-Chen Hsu, Chih-Yu Chang

**Affiliations:** Department of Mechanical Engineering, National Yunlin University of Science and Technology, Douliu 64002, Taiwan

**Keywords:** acoustofluidics, acoustic separation, acoustic aggregation, Lamb wave

## Abstract

In this study, we realize acoustic aggregation and separation of microparticles in fluid channels driven by standing Lamb waves of a 300-μm-thick double-side polished lithium-niobate (LiNbO_3_) plate. We demonstrate that the counter-propagating lowest-order antisymmetric and symmetric Lamb modes can be excited by double interdigitated transducers on the LiNbO_3_ plate to produce interfacial coupling with the fluid in channels. Consequently, the solid–fluid coupling generates radiative acoustic pressure and streaming fields to actuate controlled acoustophoretic motion of particles by means of acoustic radiation and Stokes drag forces. We conducted finite-element simulations based on the acoustic perturbation theory with full-wave modeling to tailor the acoustic and streaming fields in the channels driven by the standing Lamb waves. As a result, the acoustic process and the mechanism of particle aggregation and separation were elucidated. Experiments on acoustic manipulation of particles in channels validate the capability of aggregation and separation by the designed devices. It is observed that strong streaming dominates the particle aggregation while the acoustic radiation force differentially expels particles with different sizes from pressure antinodes to achieve continuous particle separation. This study paves the way for Lamb-wave acoustofluidics and may trigger more innovative acoustofluidic systems driven by Lamb waves and other manipulating approaches incorporated on a thin-plate platform.

## 1. Introduction

Microscale particle and cell separation is a critical technique in medical, biological, and biochemical applications [1,2]. During the past decade, a variety of approaches of particle separation has been developed using microfluidic strategies [3]. These approaches can be divided into passive and active methods. On the one hand, the passive methods are typically a hydrodynamic means to achieve streamline manipulation, microstructure perturbation of flow, and deterministic lateral displacement that leads to particle size filtration and migration [4,5,6,7]. On the other hand, the active methods involve applied external fields, such as magnetic, electrical, optical, and acoustic fields, to produce effective differential forces acting on the particles [8,9,10,11,12]. Comparatively, the active methods show better flexibility and performances.

Among the active methods, acoustofluidic separation has received increasing attention in recent years because it has the intrinsic merits of being scalable, contactless, and label-free [13]. In principle, acoustofluidic separation relies on the differential effect of acoustic streaming and radiation forces acting on particles suspending in fluid. Therefore, acoustofluidic systems are capable of separating not only particles with different sizes but also particles with different mechanical properties (i.e., density and compressibility). An acoustofluidic device is typically composed of a piezoelectric element to generate acoustic waves via electrical excitation and a microfluidic domain which carries particles to be manipulated. According to the mode of wave production by the piezoelectric element, acoustofluidic devices can be mainly classified into bulk acoustic wave (BAW)- and surface acoustic wave (SAW)-based platforms [14,15,16]. The former usually employs PZT transducers to launch bulk-borne waves into a resonant microchannel or cavity with sound-hard boundaries. However, the SAW-based platform employs interdigitated transducers (IDTs) on the surface of a piezoelectric substrate to launch surface bound waves to couple the liquid at the solid–fluid interface. The interfacial coupling makes the SAW-based platform efficient and flexible in shaping the acoustic pressure and streaming fields in microfluid channels or cavities. Thus, the SAW-based platform is actively tunable using frequency, phase, or amplitude modulation of the excited SAW field to achieve sophisticated and multidimensional acoustic tweezering [17,18,19,20,21].

The SAWs utilized in acoustofluidic systems are usually the Rayleigh mode supported by a thick substrate. However, when the thickness of the substrate is reduced, such that it is thinner than the acoustic wavelength, multiple dispersive acoustic plate modes, the so-called Lamb-wave modes, are allowed [22,23]. These plate thickness bound modes provide multiple distinct acoustic frequencies which are promising for acoustofluidic manipulation of particles at various sizes without changing the excitation configuration. Table 1 compares the differences between the SAW and Lamb wave characteristics. With the distinct characteristics of the Lamb wave form SAW, Lamb waves are able to enhance microfluidic mixing via multimode stirring [24], to change the aggregation pattern of microparticles in a microfluidic chamber with different Lamb modes [25], and to switch the moving trajectory of a microparticle in a channel by changing the driving mode from the SAW mode to Lamb mode [26]. Similar to the SAW, Lamb waves may also have highly defined mode profiles associated with the wavelength for shaping the acoustic pressure and streaming fields in the fluid domain via the interfacial coupling. We have previously demonstrated the capability of acoustic pattering of microparticles in a fluid chamber by Lamb waves [25]. In the present study, we demonstrate that, by forming standing Lamb waves across a microfluid channel, continuous particle aggregation and separation in a channel flow can be achieved.

The remainder of this paper is structured as follows. In Section 2, we describe the working principle of Lamb-wave acoustofluidic manipulations and a numerical method for the related simulations. We also describe the configuration of the devices designed and fabricated to aggregate and separate particles in channels. In Section 3, we present and analyze the numerical and experimental results. The results of Lamb-wave excitations, acoustophoretic effects, and particle aggregation and separation are discussed, accordingly. Lastly, we conclude this work with an outlook in Section 4.

## 2. Materials and Methods

### 2.1. Working Principle and Numerical Approach

As shown in Figure 1, the investigated Lamb-wave acoustofluidic device consists of a straight microchannel with a rectangular cross-section made of PDMS and two IDTs on a thin piezoelectric substrate (128°*Y*-*X* LiNbO_3_) of thickness *h*_LN_. The microchannel is centrally positioned between the two IDTs. The IDTs comprise periodically arranged finger electrodes of alternating width and spacing. The pitch is denoted as *p*_IDT_. When the piezoelectric substrate is sufficiently thin (referred to as a plate), compared with the IDT pitch, both the substrate top and bottom surfaces are effective to couple the dynamic elastic deformation generated by the IDTs with applying an alternating voltage at suitable frequencies because of the inverse piezoelectric effect. Accordingly, counter-propagating Lamb waves can be simultaneously launched by the two IDTs to form plate-bound standing waves.

As soon as the Lamb waves meet the microchannel filled with fluid, the acoustic energy is coupled directly from the solid plate into the fluid along the solid–fluid interface. The coupling results in an interfacial standing acoustic field that upwardly radiates leaky longitudinal acoustic waves into the liquid bulk. Furthermore, the viscous damping of the leaky longitudinal acoustic wave induces the acoustic streaming effect near the boundary layer of the fluid side to generate vortex flows. Consequently, the standing acoustic pressure and acoustic streaming flow actuate the acoustophoretic motion of particles by means of acoustic radiation and Stokes drag forces, respectively.

For the numerical approach, we adopt a full cross-sectional model of the acoustofluidic device by means of the finite element (FE) method, as schematically shown in Figure 1b. Hence, the excitation of Lamb-wave modes and the influence of the corresponding displacement profile and solid–fluid coupling on the resulting acoustic pressure field and streaming flow were taken into account. The overall system of the FE algorithm integrates the solid and fluid domains of the device model with continuous physical conditions along their interface. The governing equations implemented in the FE calculations involve an elastodynamic formulation with a piezoelectric effect for the solid domain and an acoustofluidic perturbation theory for the fluid domain [27,28,29,30].

In the piezoelectric solid domain, the mechanical displacement *u_j_* and the electric displacement *D_i_* of elastic-wave propagation are governed by Cauchy’s equation and Gauss’s law, respectively [31]:(1)$$\begin{array}{l} \partial_{i}T_{ij}=-\rho_{s}\omega^{2}u_{j},\\ \partial_{i}D_{i}^{}=0, \end{array}$$
where *ρ*_s_ is the mass density of the solid, *ω* is the acoustic angular frequency, and *T_ij_* is the elastic stress tensor. The stress and electric displacement obey the piezoelectric constitutive laws [32]:(2)$$\begin{array}{l} T_{ij}=c_{ijkl}^{}S_{kl}-e_{kij}E_{k},\\ D_{i}=e_{ikl}S_{kl}+\varepsilon_{0}^{}\varepsilon_{ik}^{}E_{k}, \end{array}$$
where *S_kl_* and *E_i_* are the elastic strain tensor and the electric field, respectively; *c_ijkl_* is the elastic stiffness; *e_kij_* is the piezoelectric coefficient; *ε_ik_* is the dielectric constant; and *ε*_0_ is the vacuum permittivity. The strain is related to the displacement by $S_{kl}=\left(\partial_{l}u_{k}+\partial_{k}u_{l}\right)/2$, and the electric field is related to the electric potential by $E_{i}=-\partial_{i}\varphi$.

The electrical excitation of Lamb waves can be modeled by imposing periodic surface electric potentials on the electrodes of the IDTs on the top surface of the plate as follows [27,28]: (3)$$\{\begin{array}{c} \varphi_{c}=V_{0}\;e^{i\omega t},\;\mathrm{on}\;\mathrm{the}\;\mathrm{charged}\;\mathrm{electrodes},\\ \varphi_{g}=0,\;\;\;\;\;\mathrm{on}\;\mathrm{the}\;\mathrm{ground}\;\mathrm{electrodes}. \end{array}$$

Along the non-electrode surfaces of the plate that have contact with air, the stress-free and open-circuit boundary conditions are specified:(4)$$\{\begin{array}{c} \;T_{ij}^{}n_{j}=0,\\ \;\;D_{j}^{}n_{j}=0, \end{array}$$
where *n_j_* is the unit vector outwardly normal to the boundary.

In the fluid domain, the equations that govern the acoustic and streaming flow fields are generated by the asymptotic expansion of the conservation laws of mass and momentum. The velocity *v*_1*j*_ and the pressure *p*_1_ of the acoustic field are described by the first-order equations as follows [28,30]:(5)$$\begin{array}{l} -i\omega \rho_{1}+\rho_{0}\partial_{j}v_{1j}=0,\\ -i\omega \rho_{0}v_{1j}=-\partial_{i}p_{1}\delta_{ij}+\eta \partial_{k}\partial_{k}v_{1j}+\left(\eta_{B}+\frac{1}{3}\eta \right)\partial_{j}\partial_{l}v_{1l}, \end{array}$$
where *ρ*_0_ and *ρ*_1_ are the undisturbed and first-order densities of the fluid, respectively, and *η* and *η_B_* are the shear and bulk viscosities, respectively. Here, the first-order fluctuations of the density and pressure can be related by the constitutive relation *ρ*_1_*c*_0_^2^
*= γp*_1_, where *c*_0_ and *γ* are the speed of sound and the specific heat capacity of the fluid. Furthermore, the acoustic streaming flow can be described by the time-averaged second-order equations as follows [28,30]: (6)$$\begin{array}{l} \langle \rho_{0}\partial_{j}v_{2j}\rangle =-\partial_{j}\langle \rho_{1}v_{1j}\rangle ,\\ \langle \rho_{1}{\dot{v}}_{1j}\rangle +\rho_{0}\langle v_{1k}\partial_{k}v_{1j}\rangle =-\partial_{i}\langle p_{2}\delta_{ij}\rangle +\eta \partial_{k}\partial_{k}\langle v_{2j}\rangle +\left(\eta_{B}+\frac{1}{3}\eta \right)\partial_{j}\partial_{l}\langle v_{2l}\rangle , \end{array}$$
where *v*_2*j*_ and *p*_2_ are the second-order velocity and pressure, respectively. In Equation (6), the source terms to actuate the streaming velocity *v*_2*j*_ result from nonlinear products of the first-order quantities. 

The first- and second-order fields are obtained through a two-step solution procedure using commercial FE solver package COMSOL Multiphysics [33]. First, the acoustic field of the full cross-section model can be solved using the equation system that couples Equations (1) and (5) together with the continuity of velocity and stress along the solid–fluid interface, which is given by [28]
(7)$$\{\begin{array}{c} i\omega \;u_{j}^{}=v_{1j}^{},\;\;\;\;\;\;\;\;\;\;\;\;\;\;\;\;\;\;\;\\ \;\;T_{ij}^{}n_{j}=\left(-p_{1}\delta_{ij}+\eta \left(\partial_{i}v_{1j}+\partial_{j}v_{1i}\right)+\left(\eta_{B}-\frac{2}{3}\eta \right)\partial_{k}v_{1k}\delta_{ij}\right)\;n_{j}.^{{{}^{}}^{{}^{}}} \end{array}$$

After the first-order acoustic field is obtained, the time-averaged second-order streaming field in the fluid domain can be solved accordingly using Equation (6) with the no-slip hard boundary condition.

With the first- and second-order fields, the acoustophoretic forces can be evaluated. A microparticle suspended in the fluid experiences the radiation force by scattering the acoustic waves. In the long-wavelength limit, the radiation force on the particle of diameter *d* can be approximated by [28,29]
(8)Fjrad=−πd3[κ012Re(cmp¯1∂jp1)−ρ08Re(cdv¯1k∂kv1j)],
where *κ*_0_ is the compressibility of the fluid, the bar on top denotes the complex conjugate of the quantity, and the coefficients *c*_m_ and *c*_d_ are given by
(9)$$c_{m}=1-\frac{\kappa_{p}}{\kappa_{0}},\;\;c_{d}=\frac{2\left(\rho_{p}-\rho_{0}\right)}{2\rho_{p}+\rho_{0}},$$
with *κ*_p_ and *ρ*_p_ being the compressibility and density of the particle, respectively. Simultaneously, the Stokes drag force exerted by the acoustic streaming flow on the particle is given by [28,29]
(10)$$F_{j}^{\mathrm{drag}}=3\pi \eta d\left(\langle v_{2j}\rangle -v_{j}^{p}\right),$$
where *v*^p^*_j_* is the moving velocity of the particle.

### 2.2. Device Fabrication and Experimental Procedure

The Lamb-wave device was fabricated by a standard photolithographic technology, which is conventionally applied to the fabrication of SAW devices [34]. The used piezoelectric plate is a 128°*Y*-*X* LiNbO_3_ wafer with a thickness of 300 μm. The wafer is polished on both sides to benefit regular bottom-surface reflection via the top-surface excitation of the IDTs in developing the plate-borne Lamb waves. The fabrication process involves a deposition of a thin Cu/Ti metal film on the LiNbO_3_ wafer, spin-coating of photoresist on the metal film, development of the IDT pattern on the photoresist by UV exposure, and finally chemical wet etching of the metal film to form IDT electrodes.

The microchannel to accommodate the microfluidic flow was fabricated using the replica molding method [24]. The monolithic master mold was created using stereolithographic 3D printing. After fully UV light curing to firm the master mold, a mixture of a PDMS base and curing agent (Sylgard 184, Dow Corning, Midland, MI, USA) at a ratio of 10:1 w/w was poured into the master mold to cast the microchannel structure by complying with a standard degassing and solidifying process. Finally, the PDMS replica of the microchannel was accurately combined with the Lamb-wave device to form the Lamb-wave acoustofluidic device. It is worth noting that the surface roughness could be an overall issue to influence the optical quality (transparency) of the PDMS channel as we adopt the 3D printing technique for the master mold fabrication. The issue can be improved if we adopt the conventional photolithographic and deep-etching technology on a silicon wafer to fabricate the master mold for the replica molding of the PDMS channels. However, the photolithographic and deep-etching processes increase the fabrication complexity and cost.

The Lamb waves, at the frequencies that are effective to be electrically excited in the fabricated device, were pre-determined using a network analyzer (Agilent E5061B, Keysight, Santa Rosa, CA, USA). Subsequently, in acoustophoretic experiments, the microfluidic device was actuated by injecting liquid with suspended particles into the microchannel using a syringe pump. The radio-frequency (RF) power produced at the measured Lamb-wave frequencies by a power amplifier (75A250A, Amplifier Research, Souderton, PA, USA) was delivered to the IDTs to excite the standing Lamb waves to generate the acoustophoretic forces on the particles. The resulting particle motion was recorded using a fast CCD (Prime, Photometrics, Tucson, AZ, USA) mounted on an inverted microscope (ECLIPSE Ti2-U, Nikon, Melville, NY, USA). Figure 2 shows photographs of the experimental setup, 3D-printed monolithic master mold, and Lamb-wave acoustofluidic device.

## 3. Results and Discussions

### 3.1. Excitation of Standing Lamb Modes in a LiNbO_3_ Plate

Figure 3a shows the calculated frequency dispersion relations of the Lamb-wave eigenmodes with *h*_LN_*=* 300 μm. These eigenmodes can be classified into antisymmetric (flexural, denoted as A) and symmetric (dilatational, denoted as S) modes of different orders. We focus our attention on the lowest-order fundamental modes, i.e., the A0 and S0 modes, that can easily achieve pure single-mode excitation at the wavelength *λ =* 400 μm (equivalently, *kh*_LN_/2*π =* 0.75, where *k* is the wavenumber). Accordingly, we use *p*_IDT_ *=* 400 μm in our device to correspond to the desired wavelength. Table 2 summarizes the information of the designed IDT configuration. The measured S21 spectrum of the fabricated device is shown in Figure 3b, in which several outstanding transmission peaks excited by an IDT that serves as the emitter and detected by another IDT that serves as the receiver, can be observed. Two of the peaks correspond to the A0 and S0 modes, respectively, whereas the others correspond to the higher-order Lamb-wave modes (A1, S1, A2, and S2) and the thickness-resonance mode (Th1), respectively. The frequency peaks of S21 spectrum match the theoretical eigenfrequencies of A0, S0, A1/S1, and A2/S2 Lamb-wave modes obtained from the dispersion curves in Figure 3a. Therefore, with this fingerprint, we conclude the modes corresponding to the peak frequencies of the experimental S21 spectrum. The simulated spectrum of the total displacement excited simultaneously by two IDTs that both serve as the emitters is also appended in Figure 3b, which shows peaks at the frequencies consistent with the Lamb-wave frequencies of the measured spectrum. These peaks reveal that the excited counter-propagating Lamb waves can effectively form a standing-wave field. The peak frequencies are compared in Table 3, which show small deviations between the measured and simulated values. Therefore, we conclude the validity of our simulations for the Lamb-wave device. Accordingly, the simulations are used to provide vibration displacement fields at the interested frequencies. Figure 3c illustrates the simulated displacement profiles of the standing A0 and S0 modes and their vertical displacement component *u_y_* along the plate surface at their corresponding peak frequencies. The results indicate that standing Lamb waves with clear displacement nodes and antinodes can be generated in the LiNbO_3_ plate with the double-IDT configuration.

It is known that acoustic streaming originates from the effect of a viscous boundary layer as acoustic waves transmit through a solid–fluid interface [28,29,30]. Under the no-slip condition between the deformable solid side and the fluid side along the interface, the velocity gradient in the boundary layer is significantly influenced by the motion of the solid side, especially the tangential motion. Figure 4 compares the tangential component of the surface displacement *u_x_* of the A0 and S0 modes. The result shows that the *u_x_* amplitude of the A0 mode along the top surface of the plate is considerably larger than that of the S0 mode under the same electrical excitation condition (*V*_0_ *=* 1 V). Therefore, the A0 mode is expected to be able to induce larger shear stress in the viscous boundary layer to drive stronger bulk streaming rolls in the channel.

To understand the effect of single-IDT excitation in forming the frequency peaks of plate-bound propagation in the spectrum, the displacement profiles around the IDT region were simulated, as shown in Figure 5. The profiles corresponding to the peaks denoted as Th1, A0, S0, A1/S1, and A2/S2 in Figure 3b are identified (see Table 3 for comparison of the numerical and experimental frequencies). The displacement profile of the Th1 frequency in Figure 5a shows a half-wavelength resonance along the plate thickness and localizes at the electrode region of the emitter IDT. Here, the emission of acoustic waves from the thickness-confined resonance to be detected by the receiver IDT is minor. The highly confined local resonance makes the Th1 frequency inefficient for its acoustic field to couple with the fluid in the channel. On the contrary, the displacement profiles of the A0 and S0 frequencies in Figure 5b,c show effective launch of propagating plate-bound waves associated with the A0 and S0 modes, respectively. Therefore, the double-IDT excitations at these two frequencies can be efficient to couple the resulting standing Lamb waves with the fluid in the channel. Finally, both the profiles of the A1/S1 and A2/S2 frequencies in Figure 5d,e also exhibit launch of propagating plate-bound waves but involve mixed double-mode excitations. This alters the simple sinusoidal configuration of the displacement owing to out-of-phase superposition of the two Lamb-wave eigenmodes simultaneously excited by the IDT.

### 3.2. Acoustic Field, Streaming Flow, and Acoustophoretic Forces

We then designed a Lamb-wave acoustofluidic device with the A0 and S0 modes on the basis of the discussion in the previous subsection. The simulated elastic–acoustic coupled fields on the full cross-section of the device are shown in Figure 6. The exemplified width *w* and height *h* of the channel are 600 and 60 μm, respectively. Both the standing-wave fields of the A0 and S0 modes produce three pressure antinodes and four pressure nodes along the channel width. As we set the voltage amplitude applied on the IDTs to be *V*_0_ *=* 1.0 V, the resulting maximum pressure in the fluid channel is approximately 20 kPa for the A0 mode and 18 kPa for the S0 mode. It is observed that the acoustic pressure travels upward and meets the PDMS lid. Due to the low acoustic impedance mismatch between the water and PDMS (1.5:1), the upward propagating acoustic pressure transmits mostly into the viscoelastic PDMS lid with minor reflection back to water. Therefore, we can refer to the overall acoustic field as a radiative standing acoustic field on the channel cross-section. The first-order velocity variations near the solid–fluid interface (channel bottom) are illustrated in the insets of Figure 6, which show the gradient change due to the acoustic waveform and drastic viscous damping in the boundary layer as the source to induce acoustic streaming at a slow time scale.

Figure 7 shows the simulated absolute acoustic pressure and time-averaged acoustic streaming fields on the channel cross-section induced by the standing Lamb waves of the A0 and S0 modes for different channel widths, *w =* 600 and 500 μm. Similar to the well-known SAW scheme, the standing Lamb-wave formation using the A0 or S0 mode can shape the acoustic and streaming fields in the channel where the fields exhibit highly regular profiles and spacing of pressure variations and streaming rolls, respectively. Reducing the channel width can decrease the number of pressure nodes and streaming rolls generated in the channel. Intriguingly, the streaming induced by the A0 mode is significantly stronger than that induced by the S0 mode. We attribute the stronger streaming to the higher tangential displacement component *u_x_* of the A0 mode along the solid–fluid interface. In addition, the streaming fields accompany a considerable sidewall effect on modulating the streaming rolls near the two sidewalls. It may lead to sidewall trapping of particles near the walls. The acoustic and streaming fields are then able to exert distinct acoustophoretic forces in relation to the particle size to achieve specific particle aggregation and separation through their regular spatial distributions in the channel.

We also calculated the horizontal components of the acoustic radiation force *F_x_*^rad^ and Stokes drag force *F_x_*^drag^ with respect to particles of different sizes (*d =* 2, 7, and 10 μm). The results are shown in Figure 8. We depict the force distributions along the channel width (*w =* 600 and 500 μm) at an altitude of 40 μm from the channel bottom. In obtaining the drag force, we assume that the particle velocity is zero. In the figures, it is observed that *F_x_*^rad^ and *F_x_*^drag^ act in the opposite directions, which means that they are always competing in driving the particles. For 2-μm particles, *F_x_*^rad^ is significantly smaller than *F_x_*^drag^ so that they can be better trapped at the stagnation points of the streaming rolls. For 10-μm particles, *F_x_*^rad^ is stronger to considerably actuate the particles toward the pressure nodes from the streaming drag. For 7-μm particles, *F_x_*^rad^ and *F_x_*^drag^ are close in magnitude. By comparison, the A0 mode can produce a larger drag force by stronger streaming than the S0 mode.

### 3.3. Aggregation and Separation of Microparticles

Figure 9 shows the experimental results of particle aggregation in the channel flow actuated by the standing Lamb waves of the A0 and S0 modes. The main channel flow collects two injected inlet flows of an equal flow rate (a Y-shape channel design), where one inlet flow is a 7-μm particle flow suspending in water and another inlet flow is a dyed water flow, as schematically illustrated in Figure 9a. The particles are polystyrene beads. The adopted main channel width is 500 μm. The total flow rate is 4 μL/min. The total applied RF power on the IDTs to excite the Lamb waves is 30 dBm. For using the A0 mode in Figure 9b, it is observed that the distributed particles quickly aggregate into three lines when the RF power is on (see also Appendix A). Those particles lined up along the wall are aggregated by sidewall trapping. Simultaneously, the drag force of the strong streaming rolls on the 7-μm particles competes the acoustic radiation force so that the particles close to the central line of the channel are driven to reach the dyed-water-flow side and aggregated to form a particle line. For using the S0 mode in Figure 9c, a similar process to form aggregated particle lines is observed when the RF power is on (see also Appendix A). Three aggregated particle lines are initially restricted not to cross the central line of the channel owing to weaker streaming driving. However, some particles close to the central line are gradually driven farther to the dyed-water-flow region and aggregate there.

The aggregation results of the 7-μm particles show that aggregated particle lines do not exactly locate at the simulated pressure nodes or stagnation points of the streaming rolls. Particles aggregate at pressure node of waves only when acoustic radiation force dominates, and Stokes drag force is negligible. However, according to our calculation, Figure 8 shows that the acoustic radiation force and Stokes drag force on a 7-μm particle are in the same order of magnitude, which means Stokes drag force is not negligible. In fact, the drag force induced by the acoustic streaming plays a significant role in the result of the 7-μm particle aggregation. The spacing between the aggregation lines is approximately 100 or 300 μm, which relates to the quarter wavelength of the A0 and S0 modes. This implies that the acoustic radiation force and the drag force are competing for driving the particle motion. The quarter-wavelength pattern of the streaming rolls dominates the aggregation line spacing, whereas the acoustic radiation force simultaneously expels the particles from the pressure antinodes. Moreover, complex particle–particle interaction is involved because many particles move closely, in addition to the primary acoustic radiation force and the drag force. As a consequence of these joint effects, the particles aggregate and line up at the equilibrium streamlines to move with the flow along the channel.

Subsequently, the separation of 2- and 10-μm particles is realized in a channel with three inlets (one for suspending particle flow and two for sheath flows). The configuration of the used channel design for separation is illustrated in Figure 10a. The adopted main channel width is 600 μm. The flow rates of the suspending particle flow and two sheath flows are 4 and 1 μL/min, respectively. Figure 10b,c show the separation results by using the standing Lamb waves of the A0 and S0 modes, respectively. The used total RF power is 32 dBm. When exciting the A0 mode, 2- and 10-μm particles are driven to laterally displace. Subsequently, they move along different streamlines where the size-dependent acoustophoretic forces (radiation and drag forces) are balanced so that the particles of different sizes are separated (see also Appendix A). Obviously, the separation is achieved by the differential effect of the acoustophoretic forces on the particles of different sizes, in which the 10-μm particles are subject to larger acoustic radiation force to be expelled from the pressure antinodes farther than the 2-μm particles. The manual estimation of the separation rate of the 10-μm particles separated from the 2-μm particles in Figure 10a is approximately 90%. When exciting the S0 mode, a similar differential movement between the 2- and 10-μm particles is observed, as shown in Figure 10c (see also Appendix A). However, the 2-μm particles are not highly concentrated to the aggregation lines owing to weaker streaming rolls, and thus, the separation rate is not as good as that using the A0 mode.

The difference in the streaming strength between the A0 and S0 modes leads to distinct aggregation of the 2- and 10-μm particles in separation. In the A0 mode, the strong streaming rolls enhance and stabilize the concentration of the 2-μm particles near the stagnation points (S points in Figure 8a) of the streaming rolls. However, this also tightly aggregates the particles nearby the central line of the channel by the two highly symmetrical middle streaming rolls against the expelling by the radiation force for a longer separation distance between the 2- and 10-μm particles. Consequently, the separation that occurs close to the sidewalls is more prominent in the wide channel of a 600 μm width. To break the stable central tight trapping of particles, we further investigate the particle separation in a narrower channel of a 300 μm width using the A0 mode. The configuration of the narrower channel design is illustrated in Figure 11a. Two ends of the channel are designed to expand gradually from 300 to 600 μm in width. Figure 11b,c show the result of separation at the expanding region and narrow region of the channel. The flow rates of the suspending particle flow and two sheath flows are 3 and 1 μL/min, respectively. In the narrow region of the channel where the standing A0-mode Lamb waves act, the 2-μm particles are trapped by the two middle streaming rolls but not highly aggregated into a line. In contrast, the 10-μm particles are expelled by the acoustic radiation force and separated from the particle flow to reach the sidewall (see also Appendix A). When the separated particles arrive at the expanding region, as shown in Figure 11b, the separation distance between 2- and 10-μm particles is further enlarged passively by the hydrodynamic effect of the expanding flow streamlines (see also Appendix A). The estimated separation rate of 10-μm particles separated from the 2-μm particles is higher than 90%.

## 4. Conclusions and Outlook

We have realized continuous particle aggregation and separation in microfluidic channels using standing Lamb waves. We have demonstrated that the lowest-order single-mode Lamb waves can be excited in a thin LiNbO_3_ wafer to couple the fluid and induce effective acoustophoretic forces acting on the microparticles suspended in the channel. The particles of different sizes are separated accordingly as a consequence of the differential effect of the simultaneously acting acoustic radiation and Stokes drag forces. We conducted FE simulations based on the perturbation theory with full-wave modeling to tailor the acoustic and streaming fields in the channels driven by the standing Lamb waves and elucidate the working mechanism of particle aggregation and separation. We conclude that the acoustic radiation and drag forces are competing to drive the particle motion. The strong streaming dominates the particle aggregation, whereas the acoustic radiation force differentially expels the particles with different sizes from the pressure antinodes. Consequently, particles suspending in a channel flow can be continuously separated.

It is worth noting that diverse combinations of the acoustic frequency and wavelength can be tailored with the dispersive nature of Lamb waves at different plate thicknesses. Simultaneously, multiple Lamb-wave modes with distinct dispersion relations can be chosen from. This adds extra degrees of freedom to designing acoustofluidic devices and achieving acoustic manipulations of particles in different scales via fluid-structure coupling considerations. For example, multiple Lamb-wave frequencies from low to high are available in a single device for driving resonant microfluidic cavities of different sizes, which reduces the complexity in tailoring suitable acoustic transducers. By reducing the plate thickness, the lowest antisymmetric mode exhibits a lower frequency without elongating its wavelength, which can achieve precise manipulation for particles with a larger diameter. In this study, we have shown that effective acoustophoretic forces can be generated by Lamb waves with the functions of continuous aggregation and separation in channels. Our findings pave the way for Lamb-wave acoustofluidics and may trigger more innovative acoustofluidic systems driven by Lamb waves and other manipulating approaches incorporated on a thin-plate platform.

## Figures and Tables

**Figure 1 micromachines-13-02175-f001:**
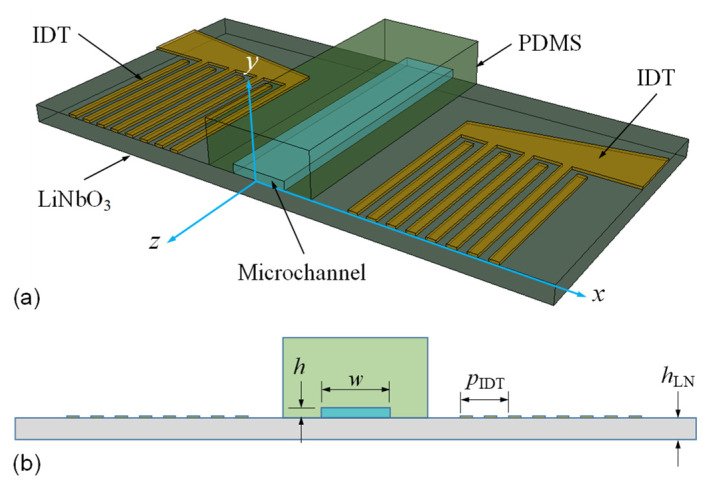
(**a**) Schematics of the sectioned structure and (**b**) the full cross-sectional model of the Lamb-wave acoustofluidic device with a rectangular water channel enclosed in PDMS. The device substrate is a 128°*Y-X* LiNbO_3_ plate with a thickness *h*_LN_, and two sets of IDTs having periodic electrode configurations are deposited on its top surface.

**Figure 2 micromachines-13-02175-f002:**
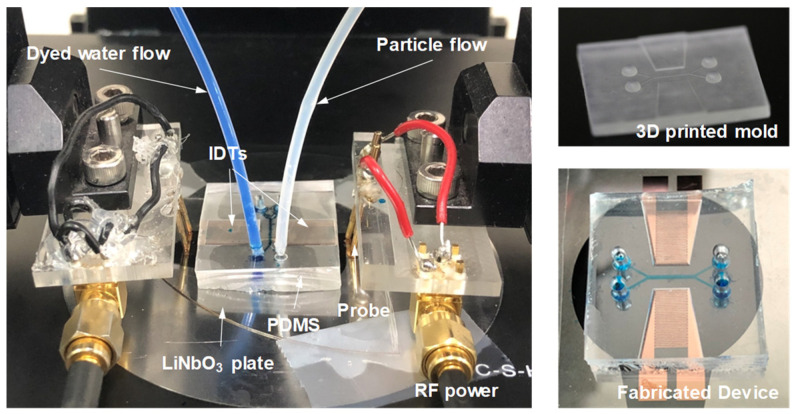
Photographs of the experimental setup, 3D-printed monolithic master mold, and Lamb-wave acoustofluidic device.

**Figure 3 micromachines-13-02175-f003:**
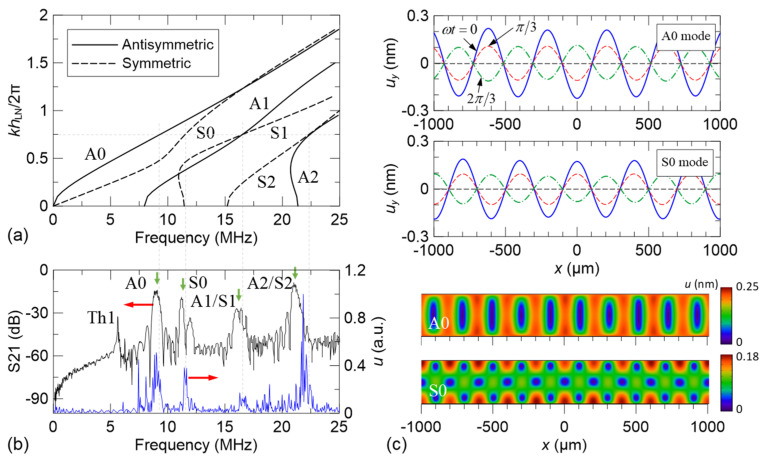
(**a**) Frequency dispersion relations of the Lamb-wave eigenmodes. (**b**) Measured S21 spectrum of the fabricated device (black curve) and simulated displacement spectrum excited simultaneously by the double IDTs (blue curve). (**c**) Vertical surface displacement component *u_y_* of the standing A0 and S0 modes at different acoustic phases (*ωt =* 0, *π*/3, and 2*π*/3) and their cross-sectional displacement profiles excited at their corresponding peak frequencies.

**Figure 4 micromachines-13-02175-f004:**
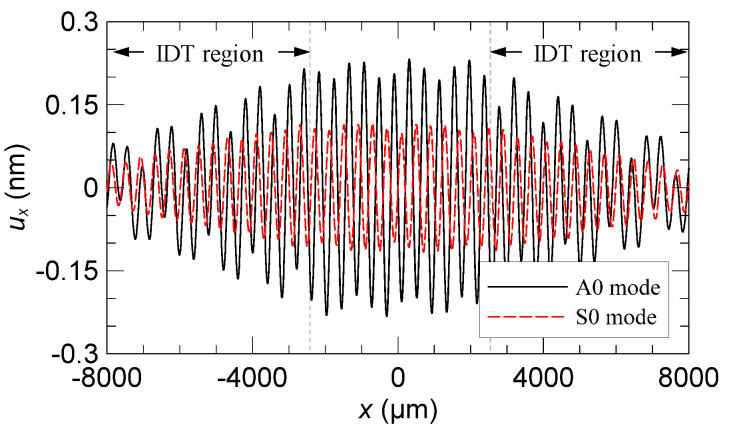
Comparison of the calculated tangential surface displacement component *u_x_* between the standing A0 and S0 modes excited by the double IDTs.

**Figure 5 micromachines-13-02175-f005:**
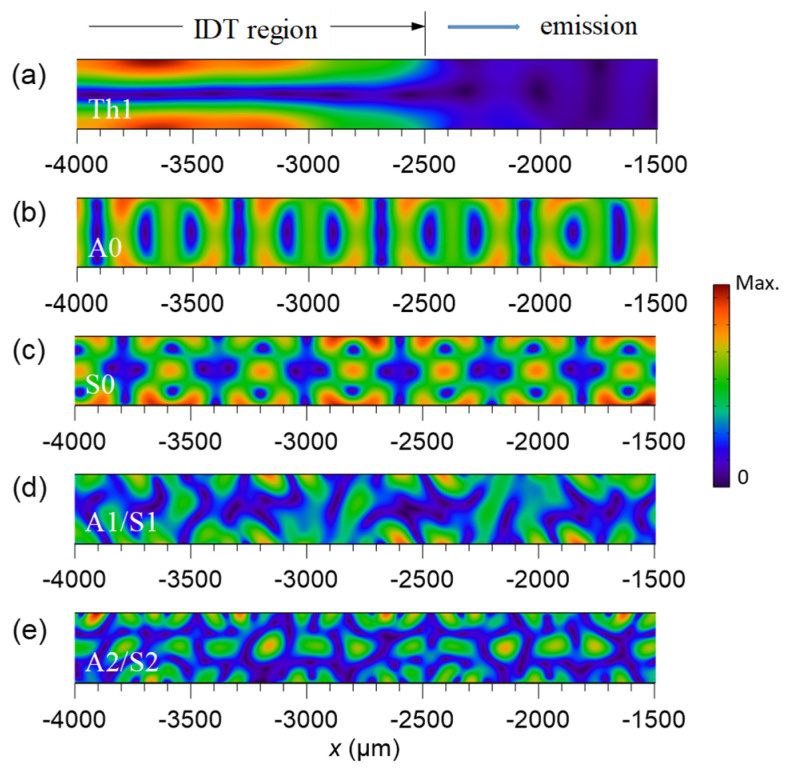
Simulated displacement profiles excited by a single IDT at the corresponding frequencies of (**a**) the thickness-resonance mode, (**b**) propagating A0 mode, (**c**) propagating S0 mode, (**d**) A1/S1 mixed modes, and (**e**) A2/S2 mixed modes.

**Figure 6 micromachines-13-02175-f006:**
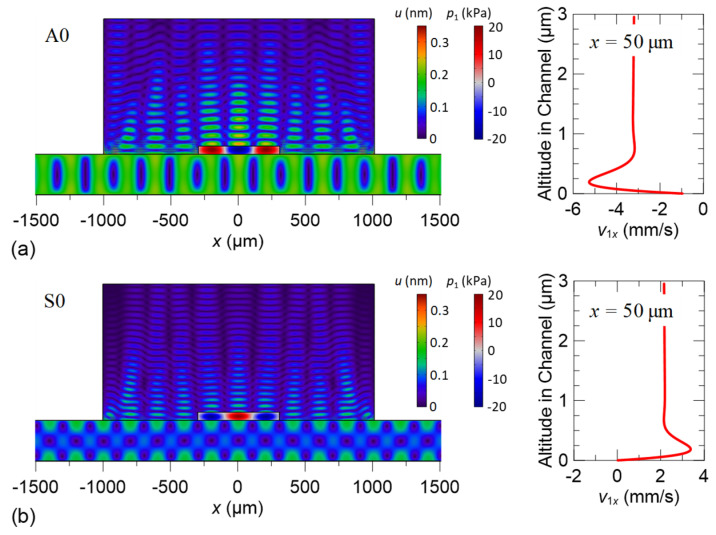
Simulated elastic–acoustic coupled fields on the full cross-section of the device and tangential velocity component *v*_1*x*_ of the acoustic fields at *x =* 50 μm near the channel bottom excited at (**a**) the A0-mode frequency and (**b**) the S0-mode frequency.

**Figure 7 micromachines-13-02175-f007:**
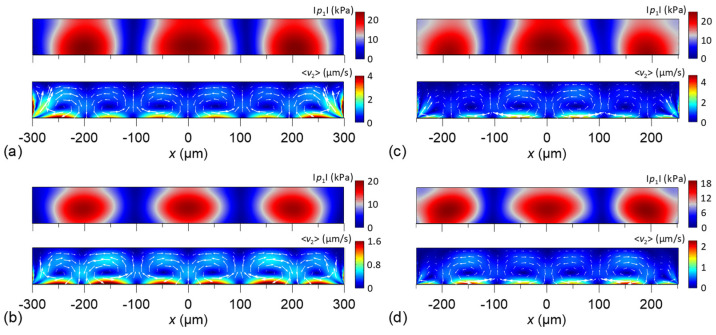
Simulated absolute acoustic pressure $\left|p_{1}\right|$ and time-averaged acoustic streaming fields $\langle v_{2}\rangle$ on the channel cross-section induced by the standing A0 and S0 modes for the channel widths *w =* 600 and 500 μm. (**a**) A0 mode, *w =* 600 μm; (**b**) S0 mode, *w =* 600 μm; (**c**) A0 mode, *w =* 500 μm; (**d**) S0 mode, *w =* 500 μm.

**Figure 8 micromachines-13-02175-f008:**
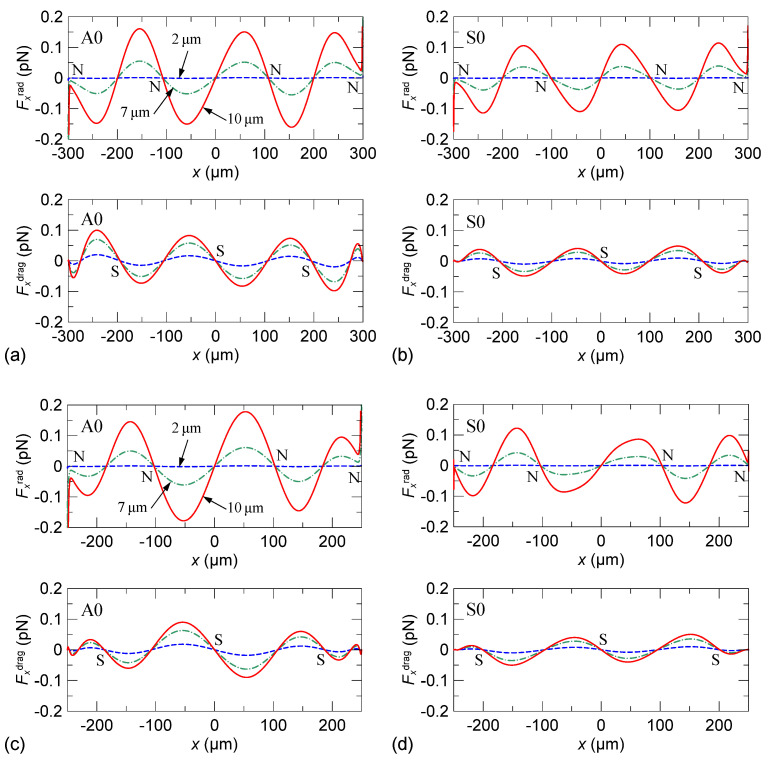
Calculated horizontal component of the acoustic radiation force *F_x_*^rad^ and Stokes drag force *F_x_*^drag^ with respect to particles of different sizes (*d =* 2, 7, and 10 μm) along the channel width at an altitude of 40 μm from the channel bottom. (**a**) A0 mode, *w =* 600 μm; (**b**) S0 mode, *w =* 600 μm; (**c**) A0 mode, *w =* 500 μm; (**d**) S0 mode, *w =* 500 μm. In the figure, N and S denote pressure nodes and streaming stagnation points, respectively.

**Figure 9 micromachines-13-02175-f009:**
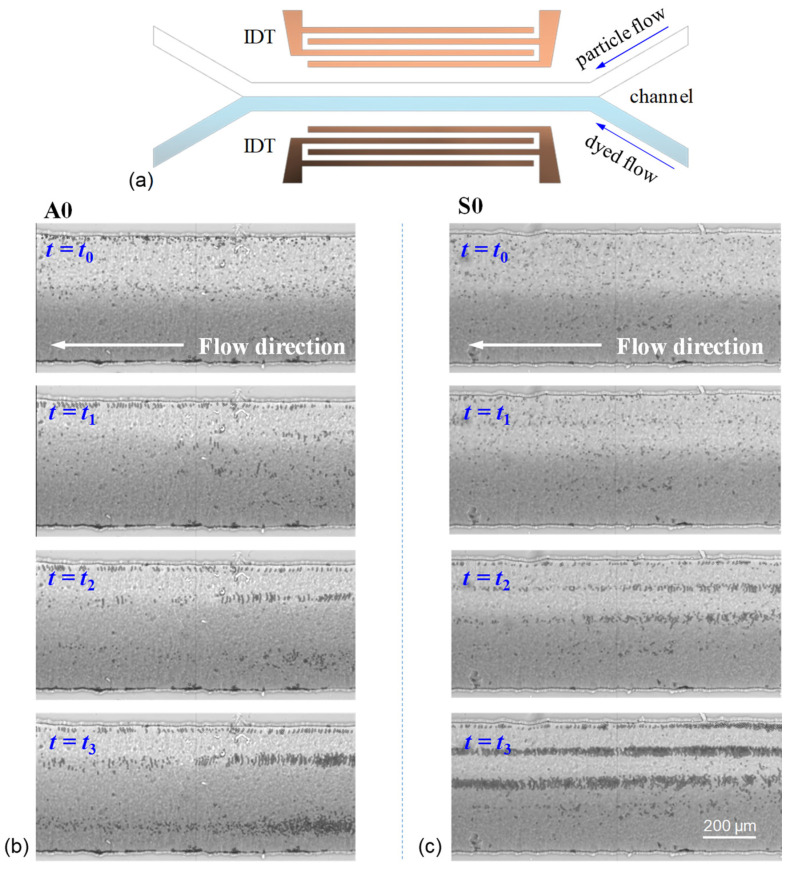
(**a**) Device layout and channel configuration with the main channel width *w =* 500 μm for acoustic aggregation. Experimental results of particle aggregation at different time *t* (*t*_0_ < *t*_1_ < *t*_2_ < *t*_3_) in the channel flow actuated by the standing Lamb waves of (**b**) the A0 mode and (**c**) S0 mode. The total volume flow rate is 4 μL/min, and the used RF power is 30 dBm.

**Figure 10 micromachines-13-02175-f010:**
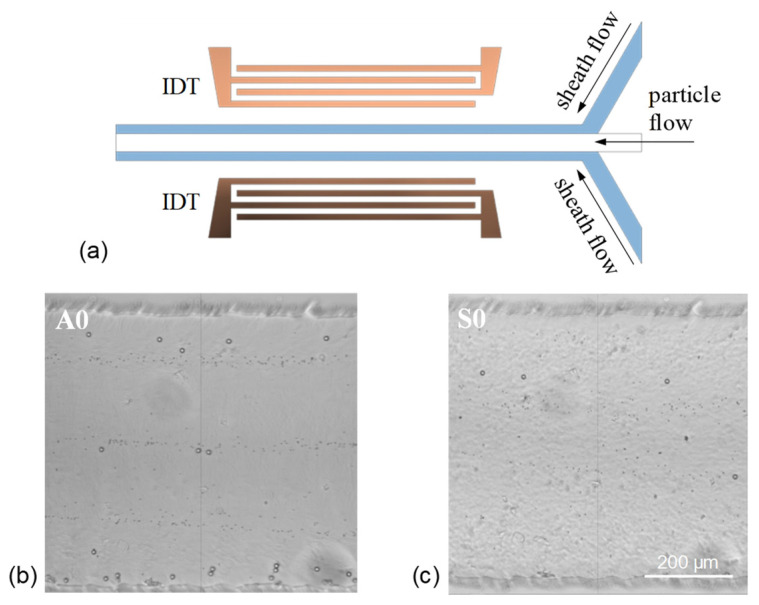
(**a**) Device layout and channel configuration with the main channel width *w =* 600 μm for acoustic separation. Experimental results of particle separation by using the standing Lamb waves of (**b**) the A0 mode and (**c**) S0 mode. The total volume flow rate is 5 μL/min, and the used RF power is 32 dBm.

**Figure 11 micromachines-13-02175-f011:**
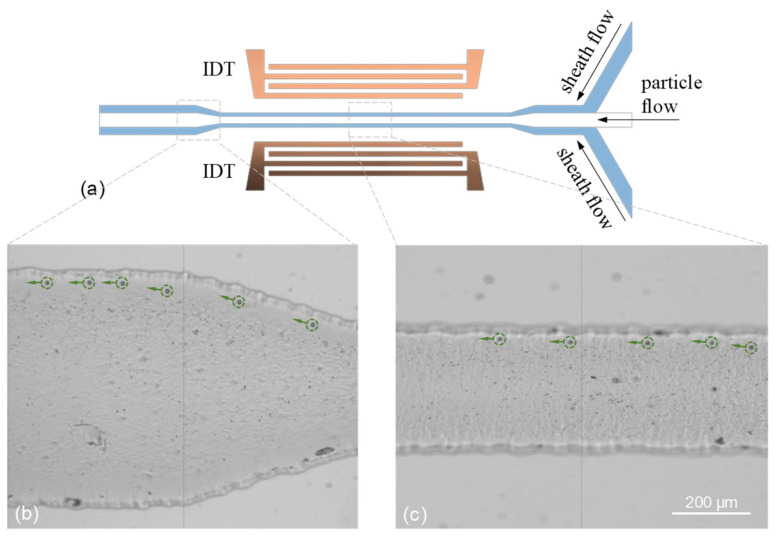
(**a**) Device layout and channel configuration, where the two ends of the channel are designed to expand gradually from 300 to 600 μm in width. Experimental results of particle separation depicted at the (**b**) expanding region and (**c**) narrow region of the channel by using the standing Lamb waves of the A0 mode. The total volume flow rate is 4 μL/min, and the used RF power is 32 dBm at the A0-mode frequency.

**Table 1 micromachines-13-02175-t001:** Comparison between the SAW and Lamb-wave characteristics.

Wave Type	SAW	Lamb Wave
Characteristics	single mode (Rayleigh mode)	multimode (antisymmetric, symmetric)
	surface localized (half-space)	plate borne (plate thickness)
	nondispersive (constant sound speed)	dispersive
	electrically excitable	electrically excitable

**Table 2 micromachines-13-02175-t002:** Design information of the substrate and IDTs.

Geometry	Value	Unit
LN substrate thickness	300	μm
IDT pitch	400	μm
IDT electrode pairs	20	
Electrode linewidth	100	μm
IDT aperture	4400	μm
Electrode thickness (Cu/Ti)	150/50	nm

**Table 3 micromachines-13-02175-t003:** Comparison of the calculated and measured frequencies excited by the IDT.

		Frequency (MHz)				
Substrate		Th1	A0	S0	A1/S1	A2/S2
128°*Y-X* LiNbO_3_	eigenfrequency	-	9.3	11.5	16.6	22.4
	calculated spectrum	6.1 ^†^	8.85	11.65	16.8	21.85
	measured spectrum	5.61	8.9	11.2	16.1	21.0

^†^ This frequency is not displayed in the calculated spectrum but is given by mode searching.

## Data Availability

The data that support the findings of this study are available within the article.

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
