# Peer review of "Continuous Particle Aggregation and Separation in Acoustofluidic Microchannels Driven by Standing Lamb Waves"

_micromachines, 2022, doi:10.3390/mi13122175_

Round 1
Reviewer 1 Report
The manuscript by Hsu et al. showed a method for continuous particle aggregation and separation in microfluidic channels using Lamb waves. They separate particles of different sizes by using the acoustic radiation force and the Stokes drag force acting on the particles simultaneously. In addition, they elucidated the working mechanism of particle separation by FE simulations. This work is written in a logical sequence, and the conclusions are appropriately supported by the data. I therefore recommend accepting this paper for publication in Micromachines after the authors addressed the following points:
(1) It would be better if figures3,4,7 and 8 align with the text.
(2) Whether or not the particle aggregation in this paper refers to the particle aggregation at the node of waves?
(3) The separation shown in the movies does not appear to be very clean, so is it possible to calculate the separation efficiency?
Author Response
Please see attached file for the author responses.

Reviewer 2 Report
In this manuscript, the authors report an acoustofluidic device based on lamb waves. By integrating standing Lamb waves of a 300-μm-thick double-side polished lithium-niobate 9 (LiNbO3) plate with the microfluidic channel, the authors developed acoustofluidic devices for the aggregation and separation of microparticles. By using simulation, the authors demonstrated the acoustic vibration and radiation force distribution of the device. The novelty of this work is the usage of standing lamb waves. The results are still preliminary. I would recommend accepting this work after addressing some points listed below:
1) It is nice to show the simulation of the acoustic field and vibration displacement of the standing lamb wave device. However, the authors need to show the experimental results. The match of both simulation and experimental results need to be discussed.
2) It is novel to use the standing lamb wave for the aggregation and separation of microparticles. However, I didn't see the advance in using the standing lamb wave instead of using the standing surface acoustic wave. The authors need to compare and show the difference. It would be nice to have a table to show the comparison.
3) The scale bars in figures 10 and 11 are missing.
Author Response

(The authors gave the same response as above.)
